# The Importance of Contextualized Psychosocial Risk Indicators in Workplace Stress Assessment: Evidence from the Healthcare Sector

**DOI:** 10.3390/ijerph18063263

**Published:** 2021-03-22

**Authors:** Luca Menghini, Cristian Balducci

**Affiliations:** Department of Psychology, University of Bologna, 40127 Bologna, Italy; cristian.balducci3@unibo.it

**Keywords:** workplace stress, psychological assessment, contextualized psychosocial risk indicators, occupation-specific, healthcare professionals, multilevel models

## Abstract

The routine assessment of workplace stress is mostly based on standardized self-report tools, including generic psychosocial risk indicators (G-PRIs) designed to fit very heterogeneous occupational sectors. However, the use “by default” of such indicators might be inadequate when they fail to characterize the specificity of the work environment; hence, the inclusion of more contextualized indicators (C-PRIs) has been recommended. We aimed at evaluating the additional contribution of three C-PRIs (Work–Family Conflict, Emotional Demands, and Excessive Demands from Patients) in predicting individual outcomes (Emotional Exhaustion, Turnover Intentions) compared to commonly used G-PRIs (e.g., Demand, Control, Support), in a sample of 787 healthcare workers involved in a routine workplace stress assessment. Multilevel hierarchical regression supported the additional contributions of C-PRIs in predicting both outcomes over G-PRIs, sex, age and shift work. More robust and consistent evidence emerged for Emotional Exhaustion, which was significantly predicted by all C-PRIs, whereas Turnover Intentions was only predicted by the C-PRI Emotional Demands. Importantly, not all G-PRIs showed a relationship with the two outcomes. Taken together, our results support the importance of including C-PRIs in workplace stress assessment carried out by organizations, which should be selected based on literature search and discussion with the stakeholders.

## 1. Introduction

Workplace stress is recognized as a widespread phenomenon that negatively affects employees’ health and wellbeing, entailing the need to be prevented and managed by organizations [1]. The 11th revision of the International Classification of Diseases included “employment conditions” (e.g., uncongenial work, stressful work schedules) among the main factors that influence health status or contact with health services. It also defined burnout as “a syndrome resulting from chronic workplace stress that has not been successfully managed” [2]. Consistent evidence of the relationship between stressful workplace conditions and both physical [3,4] and mental symptoms [5,6] has been provided. As the individual, organizational and societal costs of workplace stress have been recognized (for a review of cost-of-illness studies, see [7]), a number of international agreements and national provisions have been developed [1]. In most legal approaches, organizations are identified as responsible for monitoring employees’ levels of exposure to stressful psychosocial conditions, and preventing job-related illnesses.

The routine workplace stress assessment carried out by organizations is mostly based on standardized self-report tools designed with a degree of generality (i.e., investigating work factors that are generalizable to any job) to fit very heterogeneous occupational sectors. *Generic psychosocial risk indicators* (G-PRIs) measuring global dimensions such as job demands, job control, and social support are included in most nationally and internationally recognized instruments aiming at monitoring workplace stress [8,9,10] (for a review of dominant tools, see [11]). The use of G-PRIs is legitimized by the idea that, whereas physical hazards (exposure to noise, flammable materials, etc.) are context-specific, psychosocial hazards (excessive workload, role ambiguity, etc.) can be found in any workplace [12]. Moreover, their use allows one to develop normative data and perform cross-sectorial comparisons and benchmarking (e.g., [13,14,15]). Nevertheless, the exclusive use of a few G-PRIs, independently from the specific occupational context, might lead to an erroneous characterization of the psychosocial work environment [16,17]. The hazard identification (i.e., the identification of any significant potential source of harm that characterizes a defined group of employees) should be the first step of any risk management system [18,19], and this also applies to psychosocial hazards [20,21]. In contrast, most organizations tend to adopt conventional instruments “by default”, by implicitly assuming that their workplace is characterized by only (and by all) the considered set of G-PRIs.

Although it is recognized that certain psychosocial factors are shared by different organizational contexts, each context is characterized by occupation-specific risks. The inclusion of *contextualized psychosocial risk indicators* (C-PRI) has been recommended by several authors (e.g., [22]), and justified by theoretical frameworks such as the widely adopted Job Demands–Resources (JD–R) model [23]. The JD–R model assumes that every occupation is characterized by its own specific risk factors (job demands), which can be virtually absent in other occupations [24,25]. Occupation-specific studies supporting the JD–R assumption showed at least similar levels of covariance between wellbeing measures and both G- and C-PRIs (e.g., [26,27,28]). For instance, [29] reported a relationship between emotional demands (considered by the authors as risk peculiar to the healthcare sector) and symptoms of anxiety and depression, over and above a set of general stressors (e.g., job demand, control, support), in a sample of 256 nurses.

Despite the evidence on the role of contextualized risks, only a few studies explicitly attempted to test the additional contribution of C-PRIs. Some examples can be found within the literature on coping strategies and job resources. For instance, [30] reported higher predictive validity for a scale developed to specifically measure coping skills in mental health nurses (e.g., self-regulation and self-attitude), compared to a generic coping scale administered to the same sample. In another study [31], a scale measuring an occupation-specific job resource (camaraderie) was more consistently related to self-reported psychological health than a scale of generic emotional resources, in a sample of 547 fire-fighters. To our knowledge, only one study [32] explicitly investigated the additional contribution of C-PRIs (salary inequity and lack of human resources) over G-PRIs (demands, control, and support) in the context of workplace stress assessment, with results indicating more variance explained by the former in job satisfaction and organizational commitment, but less explained variance in psychological health, compared to the latter. Overall, the lack of systematic investigation on the unique importance of C-PRIs might be among the reasons for their scarce use by organizations in routine workplace stress assessment.

The present study is aimed at providing organizational practitioners and scholars with information on the importance of C-PRIs in workplace stress assessment. This is carried out by evaluating their additional contribution over a set of G-PRIs and demographic and occupational variables. Importantly, the data were collected during a routine psychosocial risk assessment carried out by three healthcare facilities, implying high ecological and practitioner-oriented value. The healthcare sector is particularly appropriate to our aim, as it is widely recognized as a high-risk sector, with a wide range of studies indicating the major occupation-specific psychosocial risks (for a review, see [33,34]). Among the reviewed C-PRIs, three indicators (Work–Family Conflict, Emotional Demands, Excessive Demands from Patients) were selected due to their peculiarity and based on both theoretical and statistical considerations (see Appendix A), as well as on a discussion with the stakeholders (see section below). We expected C-PRIs to significantly predict two relevant outcome variables, namely, Emotional Exhaustion and Turnover Intentions, over and above the considered set of G-PRIs. Parsimony is a critical aspect to be considered when designing a tool for the routine assessment of psychosocial risks. If C-PRIs show higher predictive power than G-PRIs, their inclusion should be considered as a priority. In contrast, if C-PRIs do not show any additional contribution, their inclusion in workplace stress assessment is not justified.

## 2. Materials and Methods

### 2.1. Participants

The data were obtained from three healthcare facilities in northern Italy between October 2015 and May 2016, during a routine psychosocial risk assessment, as required by the Italian law on workplace health and safety (Legislative Decree n. 81/2008). The assessment involved 802 employees (average sample size per facility = 263.67, SD = 146.10) who participated voluntarily. No ethical approval was sought since the study was part of a mandatory risk assessment project conducted by the facilities under the Italian health and safety law. However, the research was conducted in line with the Helsinki Declaration, as well as the Italian data protection regulation (Legislative Decree n. 196/2003). Following the methodology proposed by the Italian National Workers Compensation Authority (INAIL) [35], employees were divided into 72 homogeneous groups, defined on the basis of their facility (*N* = 3), healthcare unit (*N* = 32, e.g., Surgery, Geriatric medicine), and job profile (*N* = 7, e.g., nurses, physicians). Employees whose job profile was not related to healthcare work (i.e., 3 drivers, 12 administrative workers) were not included in the following analyses, leading to a final number of 787 employees distributed in 62 groups (mean group size = 12.69, SD = 9.06).

### 2.2. Procedure

The assessment consisted of two main steps. In the first step, an internal committee was established including employer’s delegates, safety managers, occupational physicians, and employees’ representatives. The committee was responsible for a number of tasks related to the psychosocial risk assessment, including raising awareness about the assessment among employees, and coordinating the collection of archival data. In a second step, an anonymous self-report questionnaire was administered in a paper-and-pencil modality to employees, on a voluntary basis. The questionnaires were distributed to employees via a group referent employee who was explicitly appointed for this task. Within each department involved in the project, boxes were made available to collect the filled questionnaires. Administration took place during working hours. The questionnaire included both a set of standardized G-PRIs and several C-PRIs, demographic (e.g., sex, age) and occupational variables (e.g., night shifts), as well as the two considered outcome variables.

### 2.3. Measures

#### 2.3.1. Generic Psychosocial Risk Indicators

The G-PRIs used by the INAIL methodology are derived from the United Kingdom Health Safety Executive (HSE) Indicator Tool [8,36]. All indicators were measured with items referred to the last six months, and responses were given on a 5-point Likert scale with a mixed response format using frequency (from “Never” to “Always”) and agreement (from “Strongly disagree” to “Strongly agree”).

*Demand* was measured with eight items evaluating workload, company requirements, and tasks assignments (e.g., “I have to neglect some tasks because I have too much to do”; Cronbach α = 0.81, 95% CI = [0.79, 0.83]).

*Control* was measured with six items evaluating the decisional latitude exerted by the worker on her/his job (e.g., “My job allows me to make my own decisions”; α = 0.75 [0.72, 0.77]).

*Managerial Support* was measured with five items evaluating the adequacy of encouragement, support and resources provided by the company and line managers (e.g., “I can count on the help of my supervisor when I face work-related problems”; α = 0.81 [0.79, 0.83]).

*Peer Support* was measured with four items evaluating the adequacy of encouragement, support and resources provided by the coworkers (e.g., “I receive from my coworkers the help and support I need”; α = 0. 81 [0.79, 0.83]).

*Relationships* was measured with five items on interpersonal problems or conflicts with colleagues or supervisors (e.g., “There is friction and strife among my coworkers”; α = 0.76 [0.74, 0.78]).

*Role* was measured with five items evaluating the worker’s awareness and clarity of her/his role and responsibilities within the organization (e.g., “It is clear what is expected from me for my job”; α = 0.70 [0.67, 0.73]).

*Change* was measured with three items evaluating the extent to which organizational changes are managed and communicated in the organization (e.g., “the staff is always involved on work-related changes”, α = 0.74 [0.71, 0.76]).

#### 2.3.2. Contextualized Psychosocial Risk Indicators

The G-PRIs were integrated with a set of C-PRIs designed to investigate psychosocial risks peculiar to healthcare professionals. C-PRIs were developed and partially adapted from existing scales through a process involving literature search, discussion with researchers from INAIL’s Department of Medicine, Epidemiology, and Work and Environmental Hygiene, a pilot study with 37 employees, and a workshop involving the safety managers. Among the ten proposed indicators, three C-PRIs were selected based on their peculiarity for the target occupational group, low redundancy with other indicators, and psychometric properties (see Appendix A for details). All indicators were measured with items referred to the last six months, and responses were given on a 5-point Likert scale ranging from “Never” to “Always”.

*Work–Family Conflict* was measured with three items proposed by Grzywacz and colleagues [37] to assess the frequency of experiencing work interference with family duties and activities (e.g., “My job does not allow me to spend time with my family”, α = 0.90 [0.88, 0.91]). In other words, Work–Family Conflict refers to situations where demands and responsibilities from work make it difficult to deal effectively with those characterizing the family role [38]. Such situations have been reported to be highly prevalent in healthcare professionals [37], and they are associated with several wellbeing-related outcomes, including burnout [39].

*Emotional Demands* was measured with two items included in the Copenhagen Psychosocial Questionnaire (COPSOQ) [40] evaluating the perceived emotional load involved in one’s job (e.g., “My work puts me in emotionally disturbing situations”, α = 0.72 [0.69, 0.75]). Emotional Demands (or emotional labor) can be defined as the act of expressing organizationally desired emotions during service transactions [41], and it is particularly peculiar to healthcare occupations [29,42]. 

*Excessive Demands from Patients* was measured with four items proposed by Dormann and Zapf [43] to measure service workers’ perception of highly demanding customers. To adapt the items to the healthcare context, they were rephrased to be referred to patients (e.g., “Some patients always demand special treatment”, α = 0.87 [0.86, 0.88]). Excessively demanding customers have been indicated among the social causes of burnout [43], and this factor was evaluated as prevalent and important by the panel of stakeholders.

#### 2.3.3. Individual Outcomes

The two following self-reported variables were considered as outcomes at the individual level:

*Emotional Exhaustion* was measured with five items from the Copenhagen Burnout Inventory [10,44] measuring work-related burnout (e.g., “I feel worn out at the end of the working day”) through a 5-point Likert Scale ranging from 1 = “Never” to 5 = “Always” (α = 0.92 [0.91, 0.92]).

*Turnover Intentions* was measured using two items proposed by Kelloway and colleagues [45] (e.g., “I often think to quit from my organization”) using a 5-point Likert response scale from 1 = “Strongly disagree” to 5 = “Strongly agree” (α = 0.85 [0.84, 0.87]).

### 2.4. Data Analysis

Due to the multilevel data structure (i.e., individual responses nested into groups of co-workers sharing common work features), the analyses were performed using linear mixed-effects regression (LMER), allowing us to control for the groups’ variability around the sample intercept. Intraclass correlation coefficients (ICCs) were used to estimate the amount of variance located at the group and the individual level, respectively. Regardless of the ICC value, LMER has been recommended in any case the data present a nested structure, in order to prevent too conservative Type II Errors for level-1 (individual) parameters [46]. Among the possible alterative criteria to account for the group level, we arbitrarily chose to establish groups based on the INAIL’s methodology [35] (see Section 2.1) since it was used in each facility for planning and implementing the assessment.

The additional contribution of C-PRIs was evaluated using a three-step hierarchical regression for each outcome. An initial model including only the groups’ variability around the intercept and the random error (i.e., null model with no fixed effects) was compared with three alternative models. First, a model including sex (female/male), age group (“Under 30”, “31–50”, “Over 50”), and the presence of night shift (yes/no). Secondly, the seven G-PRIs were included in model 1. Thirdly, the three selected C-PRIs were included in model 2. For each outcome, the models were compared in terms of explained variance (as indicated by the marginal coefficient of determination, R^2^), using the likelihood ratio test (LRT; testing the difference between two models in terms of likelihood), and the Akaike information criterion (AIC; comparing a set of models in terms of likelihood and parsimony) [47,48]. The model showing a substantial R^2^ increase, a significant LRT (i.e., *p* < 0.05), and the lowest AIC was considered as the best model. AIC weights were also computed and interpreted as the probability that the model will make the best prediction for new data given the set of considered models [49].

Model comparison was preceded by descriptive and correlational analysis, evaluation of LMER assumptions, and assessment of influential cases [50]. Moreover, given the potential overlap among some indicators, we assessed the risk of multicollinearity (see Appendix A for details). All analyses were performed using R 3.5.1 (R Foundation for Statistical Computing, Vienna, Austria) [51].

## 3. Results

The average response rate per job profile ranged from 67.16% for the nursing staff to 69.89% for physicians, with the highest response rates for the emergency care (86.83%) and geriatric unit (80.20%). Respondents were mainly women (*N* = 553, 70.27%), Italian (*N* = 767, 97.46%), and aged between 31 and 50 years (*N* = 503, 63.91%), with 26.18% of the sample being 51 years old or higher and only 8.13% being younger than 30. The most represented job positions were nurses (*N* = 395, 50.19%), physicians, physicists, biologists, or chemicals (*N* = 188, 23.89%), and sociomedical operators (*N* = 125, 15.88%). On average, respondents have been employed in the same facility for 14.29 years (SD = 9.89), and in the same ward for 10.35 years (SD = 8.08), with most of them being employed with a permanent contract (*N* = 736, 93.52%) including night shifts (*N* = 571, 72.55%). In contrast, the contract included extended work availability only for the 26.30% of the sample. Missing data ranged from 4 (0.51%) to 9 (1.14%) in all considered variables, except for Excessive Demands from Patients (87 missing responses, 11.05%).

Table 1 shows descriptive statistics of and first-order correlations between the considered variables. On average, the sample reported relatively high scores in job resources such as Role, Peer Support, Managerial Support, and Control. In contrast, relatively low average scores were reported for G-RPIs of job stressors (i.e., Relationships and Demand), and both individual outcomes, whereas average C-PRIs scores were all above the central point of the response scale (i.e., 3). Higher variability was observed for individual outcomes (especially Turnover Intentions) and Work–Family Conflict. ICCs reported in Table 1 range from 0.05 to 0.30, suggesting the most substantial part of the observed variance in all variables to be located at the individual level. Excessive Demands from Patients was the indicator showing the highest group-level variability.

Emotional Exhaustion and Turnover Intentions showed relatively high correlations with all PRIs, ranging from 0.24 to 0.57 and from 0.21 to 0.41 in absolute value, respectively. All correlations were in the expected direction, with job stressors (e.g., Demands, Relationships, Work–Family Conflict) being positively correlated, and job resources (e.g., Support, Change) being negatively correlated with the two outcomes. Role and Excessive Demands from Patients were the PRIs showing the lowest correlation with outcome variables. On average, Turnover Intentions showed lower correlations with C-PRIs than with G-PRIs. Critically, most of the included risk indicators were mutually correlated, especially in the case of Demands and Relationships, and Work–Family Conflict and Emotional Demand (both correlated with *r* = 0.49, *p* < 0.001), introducing a risk of multicollinearity in the following models. The two outcome variables were also highly correlated.

Table 2 and Table 3 reports the results of the model comparison and the hierarchical LMER performed for Emotional Exhaustion and Turnover Intentions, respectively. Due to missing data, all regression models were fitted on 667 observations. For both outcomes, model 3 (including both G-PRIs and C-PRIs) was selected as the best model in terms of both AIC and LRT, supporting the additional contribution of contextualized indicators. This was more evident for Emotional Exhaustion, where all C-PRIs showed a significant positive relationship, leading to an overall increase of 9% in the explained variance. In contrast, the additional contribution of C-PRIs was less consistent for Turnover Intentions, with only Emotional Demands showing a significant relationship, leading to an R^2^ increase of only 0.01.

Among the considered G-PRIs, only Demand and Peer Support were significantly associated with Emotional Exhaustion, with higher Demand and lower Peer Support being associated with higher exhaustion (see Table 2). Managerial Support and Relationships were also associated with Emotional Exhaustion in step 2, but the inclusion of C-PRIs (step 3) led to a decrease in the parameters estimated for these predictors. Higher Turnover Intentions were significantly predicted by lower Managerial Support and Change, and by higher Relationships, and these association remained significant also in step 3 (see Table 3). The inclusion of G-PRIs implied a higher increase in the explained variance for Emotional Exhaustion (39%) than for Turnover Intentions (23%), compared to step 1. Among the demographic predictors, only respondents’ age was significantly associated with Emotional Exhaustion, with respondents aged from 31 to 50 reporting higher exhaustion than respondents aged 30 or less. None of the demographic variables were associated with Turnover Intentions, and none of the outcome variables showed a relationship with night shifts.

A risk of multicollinearity and the presence of influential cases (i.e., cases that strongly determine the size of the estimated parameters) were detected in both sets of models (see Appendix A for details). In brief, multicollinearity was mainly due to strongly associated G-PRIs (e.g., Managerial Support and Relationships), whose removal did not substantially affect the results. In contrast, the parameters estimated for both G- (Managerial and Peer Support, Role and Change) and C-PRIs (Emotional Demand and Work–Family Conflict, but only in the model predicting Turnover Intentions) showed to be substantially influenced by the inclusion of specific groups of co-workers, questioning the reliability of their role in predicting this outcome.

## 4. Discussion

Our study aimed to investigate the importance of including C-PRIs in workplace stress assessment. This was carried out by evaluating their additional contribution over a set of standard G-PRIs in predicting two individual outcomes relevant for the healthcare sector. Although previous studies have supported the importance of occupation-specific risks [29,32], this was one of the first studies explicitly aiming at evaluating the additional contribution of contextualized indicators using data collected in a routine psychosocial risk assessment. This is particularly important for questioning the general tendency of using standardized measurement tools. Indeed, our results may help organizations in identifying the “trade-off” between being able to compare their psychosocial workplace with that of similar companies, and the need of employing parsimonious tools focused on the most relevant factors to be assessed. Importantly, the considered C-PRIs were selected both through theoretical and statistical considerations, and by involving the stakeholders of the healthcare facilities under assessment, coherently with the “control cycle” recommended for any risk management system [21].

Our results consistently supported the additional contribution of C-PRIs for both considered outcomes. The most reliable evidence was found for Emotional Exhaustion, the core component of burnout and a widely focused outcome of workplace stress [10,52], which showed a significant and consistent relationship with all three considered C-PRIs. In contrast, the additional contribution of C-PRIs was less evident for Turnover Intentions, which was more strongly correlated with G-PRIs, and only predicted by the C-PRI Emotional Demands. Moreover, the relationship between Turnover Intentions and C-PRIs was inconsistent across the sample: with the exclusion of influential cases, Turnover Intentions showed an increased association with Work–Family Conflict, and a decreased association with Emotional Demands. Coherently, C-PRIs showed a higher increase in explained variance for Emotional Exhaustion than for Turnover Intentions. This difference might be partially due to temporal proximity, with burnout being a more proximal outcome of perceived workplace stress compared to Turnover Intentions. Thus, in cross-sectional investigations such as ours, it may be easier to observe the potential impact of distressing working conditions (both generic and occupation-specific) on Emotional Exhaustion rather than on more distal outcomes [53].

Our results also provide valuable information regarding the relative importance of G-PRIs in the context of occupation-specific workplace stress assessment. As in the case of other widely used standardized tools [11], the set of indicators included in the HSE Indicator Tool [8] is designed to fit different occupational contexts by including the factors thought to be most relevant for work-related psychosocial wellbeing. Nevertheless, only a few of the seven included G-PRIs were consistently associated to the considered outcomes, with Emotional Exhaustion being only predicted by Demand and Peer Support, and Turnover Intentions showing a consistent association only with Relationships (i.e., the role of Managerial Support and Change was partially due to influential cases). Moreover, both sets of models were associated with a risk of multicollinearity that was mainly due to strong relationships observed among G-PRIs. This methodological concern (see Appendix A for details) suggests a degree of redundancy among standardized indicators, questioning their usefulness for occupation-specific assessment, given the need of short and parsimonious tools.

Finally, the present study adds to the existing literature on the most relevant psychosocial factors peculiar to healthcare professionals’ (and particularly, nurses’) health and wellbeing (for reviews, see [33,34]). For instance, our results are in line with previous studies highlighting the relationship between nurses’ wellbeing (e.g., emotional exhaustion, chronic fatigue, psychosomatic complaints) and both cognitive [54,55,56] and emotional job demands [57], work–family conflicts [37,39,54], and excessively demanding, difficult or hostile patients [54,58]. Patient-related social stressors have been highlighted among the most relevant factors to be considered when examining nurses’ wellbeing (see [59]), with emotional exhaustion being found predicted by perceived hostility from patients [54], verbal abuse from both patients and visitors [60], and by the frequency of interactions with difficult patients [58]. Further potentially highly relevant psychosocial stressors (and thus, potentially relevant C-PRIs) that were not considered in our study include emotional labor (e.g., surface or deep acting with patients) [58], perceived emotional display rules [61], and other relational work characteristics, such as perceived social impact and social worth [62]. All these factors should be considered when planning a workplace stress assessment in the healthcare sector, as they are likely to represent reliable sources of information about the actual state of workers’ exposure to psychosocial hazards.

The results of the present study should be interpreted in the light of some limitations. Firstly, only self-report data were used, implying that the observed relationships might be partially due to common method bias [63]. Although the INAIL’s methodology [35] provides group-level objective indicators (e.g., sick leave absences, work-related injuries), such information was available only for a small number of groups (26.42%), and considered insufficient for testing the models of interest. Notably, previous occupation-specific studies (e.g., [64]) suggested that C-PRIs (e.g., excessive noise, poor visibility, and dusty conditions) can be more predictive of objective safety indicators (e.g., number of accidents and injuries) than G-PRIs (e.g., workload, unclear role and responsibility). Secondly, the analyses were performed considering different job profiles employed in the healthcare sector. Although such profiles share several aspects of work content and context, they might be characterized by peculiar aspects influencing the established relationships. Thirdly, Emotional Demands and Turnover Intentions were measured using scales consisting of only two items, with lower reliability compared to other measures. The lack of reliability might have played a role in the inconsistencies observed for these indicators. Fourthly, due to the practitioner-oriented nature and aims of our study, we intentionally focused our analyses on direct effects only. Buffering effects (such as, for example, that of job control and social support on the strain potential of job demand) are rarely (if ever) considered in routine workplace assessments carried out by organizations. Yet, they are theoretically (and practically) as relevant as direct effects (see [65]), and their inclusion in our analyses might have led to additional explained variance in the focused outcomes. Finally, our study focused on the healthcare sector, and the results might not be generalizable to other occupational sectors. Future studies should replicate our analyses by including further and more reliable C-PRIs (e.g., measured with more items) and testing different sets of standard G-PRIs on alternative and more specific job profiles. Multimethod approaches including both self-report and objective indicators of stressors (e.g., work hours) and strain (e.g., sick leave absences, blood pressure), in addition to longitudinal studies accounting for the temporal proximity of the considered variables, are also recommended.

## 5. Conclusions

In conclusion, our study supports the importance of including occupation-specific indicators in the assessment of workplace stress, as indicated by the additional contribution of C-PRIs over G-PRIs in predicting relevant outcomes. Two main practical implications may be inferred from our results. Firstly, it is crucial to carefully select the indicators to be included, as the hazard identification should be the first step of any risk management process. Literature search and discussion with the stakeholders are the main strategies to be used in this step, and a priority to contextualized indicators is recommended. Secondly, standardized measurement tools should not be used “by default”, as some of them might be irrelevant or unreliable for certain occupational contexts, and their inclusion might mislead the assessment. Contextualized and possibly tailor-made indicators should be preferred, potentially resulting in tailor-made interventions for reducing the exposure to psychosocial hazards peculiar to specific workplaces.

## Figures and Tables

**Table 1 ijerph-18-03263-t001:** Descriptive statistics and zero-order Pearson correlation coefficients between all included variables.

	Mean (SD)	ICC	1	2	3	4	5	6	7	8	9	10	11	12
1. Emotional Exhaustion	2.67 (0.91)	0.05	1	0.50 ***	0.57 ***	−0.36 ***	−0.41 ***	−0.41 ***	0.46 ***	−0.24	−0.38 ***	0.49 ***	0.48 ***	0.38 ***
2. Turnover Intentions	2.38 (1.21)	0.10		1	0.34 **	−0.31 *	−0.39 ***	−0.33 **	0.41 ***	−0.24	−0.39 ***	0.28	0.31 *	0.21
3. Demand	2.94 (0.61)	0.10			1	−0.38 ***	−0.36 ***	−0.31 *	0.49 ***	−0.23	−0.34 **	0.44 ***	0.44 ***	0.41 ***
4. Control	3.17 (0.70)	0.10				1	0.39 ***	0.34 **	−0.36 ***	0.27	0.45 ***	−0.24	−0.22	−0.19
5. Managerial Support	3.30 (0.82)	0.13					1	0.49 ***	−0.48 ***	0.41 ***	0.63 ***	−0.29	−0.26	−0.18
6. Peer Support	3.67 (0.71)	0.14						1	−0.53 ***	0.28	0.40 ***	−0.2	−0.21	−0.15
7. Relationships	2.42 (0.76)	0.10							1	−0.34 **	−0.44 ***	0.34 **	0.37 ***	0.29
8. Role	4.24 (0.56)	0.13								1	0.42 ***	−0.21	−0.1	−0.15
9. Change	3.06 (0.87)	0.11									1	−0.3	−0.28	−0.25
10. Work–Family Conflict	3.29 (0.98)	0.13										1	0.49 ***	0.27
11. Emotional Demand	3.71 (0.80)	0.10											1	0.21
12. Excessive Demands from Patients	3.10 (0.77)	0.30												1

Note: SD standard deviation; ICC, intraclass correlation coefficient; * *p* < 0.05; ** *p* < 0.01; *** *p* < 0.001. *p*-values were obtained by testing the absolute value of each correlation against 0.15 with a one-tailed *z*-test, and they were corrected for multiple testing using the Bonferroni method.

**Table 2 ijerph-18-03263-t002:** Model comparison and parameter estimates for Emotional Exhaustion.

	Step 1			Step 2			Step 3		
	Coeff. (SE)	95% CI	*t*	Coeff. (SE)	95% CI	*t*	Coeff. (SE)	95% CI	*t*
Intercept	2.30 (0.15)	2.01, 2.60	15.33	1.59 (0.39)	0.84, 2.38	4.06	0.58 (0.38)	−0.17, 1.34	1.54
Sex (men)	−0.11 (0.08)	−0.27, 0.05	−1.34	0.06 (0.06)	−0.06, 0.19	1.00	0.05 (0.06)	−0.06, 0.17	0.93
Age (31–50)	**0.46 (0.13)**	**0.21, 0.72**	**3.58**	**0.29 (0.10)**	**0.09, 0.49**	**2.83**	**0.20 (0.09)**	**0.01, 0.38**	**2.09**
Age (over 51)	**0.39 (0.14)**	**0.11, 0.67**	**2.76**	**0.23 (0.11)**	**0.01, 0.45**	**2.03**	0.17 (0.10)	−0.04, 0.37	1.59
Night shifts (yes)	−0.04 (0.09)	−0.22, 0.15	−0.40	0.02 (0.08)	−0.13, 0.17	0.24	0.09 (0.07)	−0.05, 0.23	1.26
Demands				**0.62 (0.05)**	**0.51, 0.73**	**11.47**	**0.39 (0.06)**	**0.28, 0.05**	**7.01**
Control				−0.06 (0.05)	−0.15, 0.03	−1.22	−0.03 (0.04)	−0.12, 0.05	−0.77
Managerial Support				**−0.10 (0.05)**	**−0.19, −0.01**	**−2.13**	−0.08 (0.04)	−0.16, 0.01	−1.71
Peer Support				**−0.18 (0.05)**	**−0.28, −0.09**	**−3.78**	**−0.21 (0.05)**	**−0.30, −0.12**	**−4.58**
Relationships				**0.10 (0.05)**	**0.01, 0.20**	**2.16**	0.03 (0.05)	−0.05, 0.12	0.76
Role				0.03 (0.06)	−0.08, 0.14	0.57	−0.01 (0.05)	−0.11, 0.10	−0.16
Change				−0.08 (0.04)	−0.16, 0.01	−1.82	−0.03 (0.04)	−0.11, 0.05	−0.73
Work–Family Conflict							**0.16 (0.03)**	**0.10, 0.22**	**5.23**
Emotional Demands							**0.22 (0.04)**	**0.14, 0.30**	**5.65**
Excessive Demands from Patients							**0.16 (0.04)**	**0.08, 0.23**	**4.05**
R^2^	0.02			0.41			0.50		
AIC weight	0.001			0.001			0.99		
χ^2^ (df)	14.48 (4) **			342.24 (7) ***			103.79 (3) ***		

Note: Coeff., regression coefficient, SE, standard error; CI, confidence intervals computed with bootstrap percentile method with 10,000 replicates; *t*, *t*-value associated with the estimated parameters; R^2^, marginal coefficient of determination; AIC, Akaike Information Criterion; χ^2^, likelihood ratio test statistic; df, degrees of freedom; ** *p* < 0.01; *** *p* < 0.001. Bold type indicates the significance of the estimated parameter according to the 95% CI.

**Table 3 ijerph-18-03263-t003:** Model comparison and parameter estimates for Turnover Intentions.

	Step 1			Step 2			Step 3		
	Coeff. (SE)	95% CI	*t*	Coeff. (SE)	95% CI	*t*	Coeff. (SE)	95% CI	*t*
Intercept	2.06 (0.21)	1.65, 2.47	9.94	3.16 (0.60)	1.96, 4.35	5.26	2.71 (0.62)	1.49, 3.93	4.37
Sex (men)	−0.11 (0.11)	−0.32, 0.09	−1.06	0.05 (0.10)	−0.13, 0.24	0.53	0.05 (0.1)	−0.13, 0.24	0.54
Age (31–50)	0.30 (0.17)	−0.04, 0.64	1.72	0.18 (0.15)	−0.12, 0.48	1.20	0.14 (0.15)	−0.16, 0.43	0.92
Age (over 51)	0.13 (0.19)	−0.24, 0.51	0.71	0.06 (0.17)	−0.26, 0.39	0.37	0.03 (0.17)	−0.30, 0.36	0.17
Night shifts (yes)	0.10 (0.13)	−0.17, 0.37	0.73	0.20 (0.12)	−0.03, 0.44	1.69	0.23 (0.12)	0.00, 0.47	1.94
Demands				0.13 (0.08)	−0.03, 0.29	1.57	0.02 (0.09)	−0.16, 0.20	0.21
Control				−0.09 (0.07)	−0.23, 0.05	−1.28	−0.08 (0.07)	−0.21, 0.06	−1.12
Managerial Support				**−0.19 (0.07)**	**−0.33, −0.05**	**−2.65**	**−0.18 (0.07)**	**−0.31, −0.03**	**−2.45**
Peer Support				−0.06 (0.07)	−0.21, 0.08	−0.85	−0.08 (0.07)	−0.23, 0.07	−1.04
Relationships				**0.32 (0.07)**	**0.17, 0.46**	**4.29**	**0.28 (0.07)**	**0.13, 0.42**	**3.71**
Role				−0.13 (0.09)	−0.30, 0.04	−1.47	−0.15 (0.09)	−0.32, 0.02	−1.76
Change				**−0.20 (0.07)**	**−0.33, −0.07**	**−2.97**	**−0.18 (0.07)**	**−0.31, −0.05**	**−2.64**
Work–Family Conflict							0.06 (0.05)	−0.04, 0.17	1.26
Emotional Demands							**0.14 (0.06)**	**0.02, 0.27**	**2.23**
Excessive Demands from Patients							0.05 (0.06)	−0.07, 0.18	0.82
R^2^	0.01			0.24			0.25		
AIC weight	0.001			0.10			0.90		
χ^2^ (df)	6.45 (4)			181.56 (7) ***			10.48 (3) *		

Note: Coeff., regression coefficient, SE, standard error; CI, confidence intervals computed with bootstrap percentile method with 10,000 replicates; *t, t*-value associated with the estimated parameters; R^2^, marginal coefficient of determination; AIC, Akaike Information Criterion; χ^2^, likelihood ratio test statistic; df, degrees of freedom; * *p* < 0.05; *** *p* < 0.001. Bold type indicates the significance of the estimated parameter according to the 95% CI.

## Data Availability

The data are available upon request from the first author.

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
