# Peer review of "The Importance of Contextualized Psychosocial Risk Indicators in Workplace Stress Assessment: Evidence from the Healthcare Sector"

_ijerph, 2021, doi:10.3390/ijerph18063263_

Round 1

Reviewer 1 Report

This is, in my opinion, a high-quality study examining the importance of contextualized psychosocial risk indicators in workplace stress assessment in the healthcare sector. It contributes to a better understanding about measurements tools (contextualized/tailor-made versus general/standardized). This study has many strengths. First, the research question is original and well defined. Second, the results provide an advance in current knowledge. Third, the article is well written and structured. Data are presented appropriately. Fourthly, the authors elaborate the C-PRIs with the consideration of theoretical, statistical, and more importantly by involving the stakeholders. Lastly, these results should be of interest for the readership of the Journal.  Overall, I believe that this work merits being published. However, I have minor comments and concerns that I hope will be helpful.

1-) Citations.

  • On page 2. “Balducci and colleagues” should be removed from the text.
  • On page 2. “McElfatrick and colleagues” should be removed from the text.
  • On page 2. “Tuckey and Hayward” should be removed from the text.
  • On page 10. “Amponsah-Tawiah and colleagues” should be removed from the text.

2-) On page 1. A reference should be added to support the following statement “Workplace stress is recognized as a widespread phenomenon that negatively affects employees’ health and wellbeing....”

3-) On pages 3; 5. Concerning the multilevel aspect of the study. It is mentioned on page 3 that data were obtained from three healthcare facilities. Then, it is mentioned that employees (n=802) were divided into 72 homogeneous groups. I understand that there were not a sufficient number of health care establishments in order to perform the multi-level analyzes. So groups were created. Those homogeneous group were defined based on their facility (n=3); healthcare unit (n=?); job profile (n=?). I suggest to add some information about that particular aspect of groups creation, how those groups could account for the level 2 and the necessity to consider the hierarchical structure of the data. Because on page 5, it is mentioned that individual (level 1) were nested into groups of co-workers (level 2). How is it relevant to verify the variance between groups of co-workers (level 2)?. I simply suggest to add more precision in the manuscript.

I would like to congratulate the authors for the work!

Reviewer 2 Report

Thank you for having the opportunity reviewing this paper. Yes, there are large discussions on adopting workplace assessments to the specific industry especially in healthcare. Therefore, your paper is very relevant and I only have two concerns that are unfortunately much work but I think it would improve your argumentation and your impact.

a) There are a lot of studies (especially on nurses) about the relationship on stress and strains provided by patients and EE or other outcome variables, especially in the Journal of Nursing Management, Scandinavian Journal of Caring Sciences and many others. Please include those studies for your argumentation, this will lead to a better theoretical part and especially a better discussion and therefore to an augmentation of the value of your research.

b) Please be careful with your constructs. Social support (peers, supervisors etc.) has two effects, a direct one and a buffering one. With your methodology you do not sufficiently adress the knowledge on the  effects of the selected constructs. A SEM or other methods would be better.

The rest is fine, good idea etc.

Round 2

Reviewer 2 Report

Your improvements are fine, I suggest a publication.